# Phylogenetic Review of *Acaulospora* (*Diversisporales, Glomeromycota*) and the Homoplasic Nature of Its Ornamentations

**DOI:** 10.3390/jof8090892

**Published:** 2022-08-23

**Authors:** Kássia J. G. da Silva, José A. L. Fernandes, Franco Magurno, Larissa B. A. Leandro, Bruno T. Goto, Raquel C. Theodoro

**Affiliations:** 1Programa de Pós-Graduação em Sistemática e Evolução, Centro de Biociências, Natal 59078-970, RN, Brazil; 2Centro de Biociências, Campus Central, Universidade Federal do Rio Grande do Norte, Natal 59064-741, RN, Brazil; 3Ottawa Health Research Institute (OHRI), Ottawa, ON K1H 8L6, Canada; 4Institute of Biology, Biotechnology and Environmental Protection, Faculty of Natural Sciences, University of Silesia in Katowice, Jagiellońska 28, 40-032 Katowice, Poland; 5Departamento de Botânica e Zoologia, Universidade Federal do Rio Grande do Norte, Natal 59064-741, RN, Brazil; 6Departamento de Biologia Celular e Genética, Universidade Federal do Rio Grande do Norte, Natal 59064-741, RN, Brazil

**Keywords:** biogeography, diversity, morphology

## Abstract

The genus *Acaulospora* has undergone many updates since it was first described; however, there are some missing pieces in the phylogenetic relationships among *Acaulospora* species. The present review aimed to: (i) understand the evolutionary meaning of their different spore wall ornamentations; (ii) define the best molecular marker for phylogenetic inferences, (iii) address some specific issues concerning the polyphyletic nature of *Acaulospora lacunosa* and *Acaulospora scrobiculata*, and the inclusion of *Kuklospora* species; and (iv) update the global geographical distribution of *Acaulospora* species. As such, the wall ornamentation of previously described *Acaulospora* species was reviewed and phylogenetic analyses were carried out based on ITS and SSU-ITS-LSU (nrDNA). Moreover, the already available type material of *A. sporocarpia* was inspected. According to the data obtained, temperate and tropical zones are the richest in *Acaulospora* species. We also confirmed that *A. sporocarpia* does not belong to *Acaulospora*. Furthermore, our phylogeny supported the monophyly of *Acaulospora* genus, including the *Kuklospora* species, *K. colombiana* and *K. kentinensis*. The nrDNA phylogeny presented the best resolution and revealed the homoplasic nature of many ornamentations in *Acaulospora* species, pointing out their unfeasible phylogenetic signal. This review reinforces the urgency of more molecular markers, in addition to the nrDNA sequences, for the definition of a multi-locus phylogeny.

## 1. Introduction

Arbuscular mycorrhizal fungi (AMF) belong to *Glomeromycota* [1,2,3] and form an obligatory symbiosis with approximately 80% of terrestrial plants [4] and aquatic plant communities [5,6]. This ecological relationship brings several benefits for the plant host, including greater mineral nutrition and increased resistance to biotic and abiotic stresses [4,7]. Said stresses (saline, hydrical, pathogens), as well as other general environmental factors such as temperature, precipitation and agricultural activities, affect soil conditions, influencing AMF sporulation and species establishment, shaping the geographic distribution of these fungal species [4,8,9,10,11,12,13].

The cosmopolitan distribution of *Glomeromycota* [2,14,15,16] demonstrates that the establishment of this symbiosis is both widespread [4] and ancient, and thus extremely important for the colonization of the terrestrial environment by plants [17].

Description of the *Glomeromycota* species is mostly based on morphological characterization, which is still the basis of the AMF classification [11,14,16]. However, in some cases, the same morphological characteristics can be shared by different species [16,18,19]. For example, the species *Acaulospora colombiana* and *Acaulospora koskei* both present spores with a smooth and hyaline outer wall. Additionally, the presence of non-viable or immature spores in soil samples makes the morphological identification tricky [16,20]. Therefore, the combination of molecular and morphological studies has been pointed out as the most appropriate methodology for classifying the AMF and describing their geographical distribution [15,16,20], as well as their phylogenetic relationships [19,21,22,23,24,25,26].

Among *Glomeromycota* genera, *Acaulospora* (class *Glomeromycetes*, order *Diversisporales*, family *Acaulosporaceae*), described by Gerdermann and Trappe [27], is one of the most widely distributed genera in the world, with several species having been described in recent years [13,28,29,30].

Many studies have highlighted the ability of *Acaulospora* to develop under extreme conditions, such as in highly saline soil [31,32], and also in soil contaminated with nickel [33] and arsenic [34]. This resistance to different abiotic stresses indicates the biotechnological potential of *Acaulospora* species for agricultural and bioremediation activities.

Furthermore, the genus *Acaulospora* has also been reported as the second most frequent genus in disturbed areas, with high diversity indices, mainly in South America [28,35]. Together with *Glomus* (Tul. & C. Tul.) [29,30], the genus *Acaulospora* has an essential role in the use and management of soil, especially in semiarid areas [13,28,30,36].

Taking into account the high diversity and wide distribution of the *Acaulospora* genus, as well as its relevance for biotechnological applications and area conservation, this review aims to revise the spore wall ornamentation patterns found in this genus, to update the global geographical distribution of *Acaulospora* species, and to provide a robust phylogenetic analysis based on ITS and SSU-ITS-LSU (nrDNA) sequences of the nuclear rDNA.

## 2. *Acaulospora*: A Review of Its Taxonomy, Morphology and Molecular Markers

The articles used in this review were searched using the keywords: “*Acaulospora + biogeographical*”, “*Acaulospora + description*” and “*Acaulospora + revision*”, covering articles from 1974 to 2021, in order to gather historic and the most updated information on *Acaulospora* biology. In addition to the literature review, some *Acaulospora* species were evaluated for assessment of morphological characteristics, such as ornamentation.

Eighteen voucher specimens (spores permanently mounted in PVLG and a mixture of PVLG and Melzer’s reagent (1:1, *v*/*v*) on slides) deposited at UFRN (Natal, Brazil) (Table 1) and at Oregon State University (OSC), Oregon, USA (isotypes, types and other materials) were morphologically analyzed and used as reference.

Morphological features of spores were categorized based on original species descriptions and other related references. The preparation of spores for study and photography was carried out as previously described [37]. The types of spore wall layers were defined by Błaszkowski [37] and Walker [38]. Fungi nomenclature and the authors of their descriptions were retrieved from the Index Fungorum website http://www.indexfungorum.org/AuthorsOfFungalNames.htm (accessed on 20 October 2020). The term “glomerospores” was used for spores produced by AMF, as proposed by Goto and Maia [39].

The classification and description of *Acaulospora* species is mainly based on the morphology and ontogeny of their spores. The most important characteristic in distinguishing *Acaulospora* from other *Glomeromycota* genera is the type of spore development, called acaulosporoid, in which the glomerospores develop through the transfer of the content from the sporiferous saccule, which is connected to a hypha [21,37,40]. When this saccule is released from the spore, it leaves a single scar, which is used to differentiate species with acaulosporoid development from species with other types of spore formation. However, the acaulosporoid formation is also present in the genera *Ambispora* C. Walker, Vestberg & A. Schüßler, *Archaeospora* J.B. Morton & D. Redecker, *Otospora* Oehl, Palenzuela & N. Ferrol and *Palaeospora* Oehl, Palenz., Sánchez-Castro & G. A. Silva [21,37,41,42,43,44]. When two scars are produced, the spore formation is called entrophosporoid, which is present in the genera *Entrophospora* R.N. Ames & R.W. Schneid., *Intraspora* Oehl & Sieverd., *Kuklospora* Oehl & Sieverd., *Sacculospora* Oehl, Sieverd., G.A. Silva, B.T. Goto, I.C. Sánchez & Palenzuela and *Tricispora* Oehl, Sieverd., G.A. Silva & Palenz. [21,37,45,46,47], although *Acaulospora colliculosa* Kaonongbua, J.B. Morton & Bever has been described without the register of sporiferous saccule, but with two distinct scars, suggesting an entrophosporoid spore development [48].

Generally, spores produced by *Acaulospora* species are found as free spores, rarely in aggregates or sporocarps [27,37,40], and have three walls [21]. When some ornamentation is present, it is generally observed in the second layer of the outer wall and rarely in the inner layer [49]. A granular germ layer with a “beaded” surface, which reacts to Melzer’s reagent, is also observed in the spores [27,37,40].

There are many different types of spore wall ornamentations: projections (*Acaulospora brasiliensis*, *A. colliculosa*, *A. denticulata*, *A. endographis*, *A. elegans*, *A. entreriana*, *A. flavopapillosa*, *A. ignota*, *A. pustulata*, *A. rehmii*, *A. soloidea*, *A. spinosa*, *A. spinossissima*, *A. spinulifera*, *A. tortuosa*, *A. tuberculata*, *A. walkeri* and *Kuklospora spinosa*), depressions (*Acaulospora alpina*, *A. aspera*, *A. baetica*, *A. cavernata*, *A. excavata*, *A. foveata*, *A. herrerae*, *A. kentinensis*, *A. lacunosa*, *A. minuta*, *A. nivalis*, *A. paulinae*, *A. punctata*, *A. scrobiculata*, *A. sieverdingii*, *A. taiwania*, *A. terricola* and *A. verna*) and double ornamentation (*A. bireticulata* and *A. reducta*). These ornamentations can be uniform (ellipsoidal, circular, concave round), multiform (triangular, circular, ellipsoidal, y-shaped, tooth-shaped), or irregular in shape, and densely or sparsely distributed (Table 2, Figure 1).

At the time the order *Glomerales* was proposed, encompassing all AMF [40], *Acaulospora* covered only 25 species and, when the phylum *Glomeromycota* was proposed, this number changed little, with the addition of four species [1]. In the past 20 years, seven species have been relocated to other genera and three have been transferred from other genera to *Acaulospora* (Table 3). Additionally, a significant number of new AMF species have been described as belonging to the *Acaulospora* genus, which currently comprises 60 species.

In addition to these morphological characteristics for species differentiation, molecular data have been broadly used for species descriptions and phylogenetic analyses [50]. For instance, some species, such as *A. scrobiculata* and other species with similar ornamentations (pitted surface) are only distinguished by molecular analysis, along with morphological description [21].

Forty-five (75%) *Acaulospora* species have one or more sequenced regions available (Table 4) in NCBI, EMBL, GBIF, BLOYD SYSTEMS, MaarJAM and MYCOBANK databases, most from multicopy nuclear ribosomal RNA genes, which are organized as an operon containing the sequences for the Small Subunit 18S rRNA (SSU), 5.8S and Large Subunit 28S rRNA (LSU) nrDNAs, separated by two internal transcribed spacers, ITS1 and ITS2. The sequences corresponding to the rRNA genes are more conserved than the ITS regions and, for this reason, the SSU and LSU may be used in comparing distant taxa (genus, families), while ITS is more suitable for evolutionary analysis of very closely related species or individuals from the same species [2,26,50,51]. Only eleven species have an additional gene sequenced, which include *Beta tubulin*, *Alpha tubulin*, DNA-directed RNA polymerase II subunit (*RPB1*), Chitin Synthase (*CHS*), Transcription Factor (*Ste12*), Heat-Shock Protein 60 (*HSP60*) and a Group I Intron (IGI) from the Cytochrome Oxidase 1 (*COX1*) gene (Table 4), which is still insufficient for in-depth phylogenetic study. These other sequenced genes have been proposed as secondary barcode markers, in addition to the ribosomal markers, for multi-locus approaches [22,24,26].

SSU sequences have been widely used to infer the distribution of AMF species around the world [11,14,96]. In said works, only a few species stood out as having a global distribution, such as *Acaulospora scrobiculata* [15]. Considering that this might actually reflect the conservative nature of the SSU marker, in our review, we analyzed these sequences for *Acaulospora* species in order to address their potential for species resolution. The CD-HIT [97,98,99,100] online system was used for clustering and comparison of SSU sequences (53 sequences, belonging to 28 species), with different cut-offs of similarity. With a cut-off of 94% similarity among sequences, five clusters were defined. One of these clusters encompassed 20 different species, demonstrating that this threshold was not suitable for separating species. By using a cut-off of 100% similarity, 25 clusters were formed, yet three of them included more than one species; for example, the species *A. ignota, A. baetica, A. nivalis, A. cavernata* and *A. punctata* were grouped together in the same cluster, showing the conservative nature of this sequence and its inefficiency at distinguishing among species. Another example of such inadequacy is the 98% similarity of the SSU sequence of *Acaulospora scrobiculata* with other *Acaulospora* species (*A. minuta* and *A. spinosa*).

Likewise, an unconvincing pattern of clustering was observed when testing different cut-offs for LSU sequences, which, although presenting a better species resolution than SSU, is also a conservative marker when compared to ITS sequences. This low ability for species separation was also observed in preliminary phylogenies constructed only with SSU or LSU sequences, which presented many polytomic branches as well as para or polythyletic species (Appendix A). On the other hand, when CD-HIT with cut-offs of 98 and 100% were performed for ITS sequences, a larger number of groups was formed, mostly containing sequences from the same species. However, some single species were distributed in more than one cluster. This was observed for *A. scrobiculata*, *A. spinosa, A. laevis*, *A. minuta, A. mellea* and *A. delicata*, indicating that, for these species, the intraspecific variability of ITS marker may be greater than interspecifically, or that some fungal isolates could have been erroneously identified. The best sequence clustering, for which each species formed a different single group, was achieved by using the combination of both regions (SSU/LSU plus ITS, here named nrDNA), indicating that this is the most suitable molecular marker available for understanding the evolution of *Acaulospora*.

## 3. Diversity and Distribution of *Acaulospora* Species on the Globe

All sequences, species descriptions, and geographical occurrences were obtained from the databases NCBI (National Center for Biotechnology Information [101]), EMBL (European Molecular Biology Laboratory–European Bioinformatics Institute [102]), GBIF (Global Biodiversity Information Facility [103]), BLOYD SYSTEMS (Barcode of Life Data System [104]), MaarJAM [105], MYCOBANK Database [106], from November 2018 to December 2020.

The geographic distribution of the 60 *Acaulospora* species reported so far encompasses 61 countries. The records of *Acaulospora* species richness are observed in the tropical region. Brazil (43) presents the highest occurrence (Figure 2, Table 4), followed by: India (26), Switzerland (22), Poland (18), United States (16), Argentina (16), China (15), South Korea (12), England (11), and Benin (11). Brazil is considered one of the countries with the greatest biodiversity in the world [12,16], which can be explained by its high degree of endemism, vast territorial extension and diversification of ecosystems and biomes [16].

This difference in *Acaulospora* diversity among different localities may also be explained, in part, by the scientific efforts in taxonomic studies in said regions. There is a tendency for increased occurrences in countries where this type of research is emerging, such as in Latin America [16,107], Africa and Asia [108,109,110,111]. The distribution profile of *Acaulospora* species reviewed here seems to follow this trend. For instance, there were few *Acaulospora* reports in India until 2014, when Gupta et al. [110] reviewed the diversity of the genus, recording 45% of all known *Acaulospora* species in India. The same is valid for Brazil, the richest country in *Acaulospora* species, and other AMF.

The cosmopolitan distribution of *Acaulospora* genus was also pointed out by Öpik et al. [96], who collected sequences of virtual taxa of *Glomeromycota* worldwide and demonstrated that *Acaulosporaceae* was present on all continents, with a higher frequency in Europe and South America. In relation to the climate, no virtual taxa were detected in the Boreal zone, and the richness was higher in temperate and tropical zones. A tendency was detected for some species to have a wider geographic distribution and a greater number of hosts, although subsequent studies are needed to assess this condition [52].

Cofré et al. [16] defines the distribution of AMF species in the Atlantic Forest, Cerrado and Chaco (Argentina), as a diagonal biodiversity because richness of AMF species is in agreement with that observed for other fungi and plants in said biomes. The recent review by Maia et al. [12] indicated that about 60% (192 species) of all AMF species are present in Brazilian biomes, and this bias is also observed concerning the *Acaulospora* genus, as 71.6% (43 species) of its species are found in Brazil, where the Cerrado and Atlantic Forest are the richest biomes for this genus, with 33 and 31 species recorded, respectively. Stürmer and Kemmelmeier [35] documented that, in Neotropical areas, *Acaulospora* is the most frequent genus of AMF, followed by *Glomus, Scutellospora* C. Walker & F.E. Sanders, and *Funneliformis* C. Walker & A. Schüßler, with 47, 29, 15, and 13 species recorded, respectively.

However, we cannot rule out the influence of vegetation diversity, soil condition and environmental disturbance in shaping the AMF distribution [8,9,10,11,12,13,69]. For instance, Vieira et al. [13] showed that acidity, carbon, and clay content in soil proved to be detrimental in the composition of fungi in the Brazilian semiarid region. In said work, the authors observed that *Glomus* and *Acaulospora* were more frequent in clay rich soils than *Gigaspora* Gerd. & Trappe. Interestingly, Baar et al. [4] and Sudová et al. [112] indicated that, in lake areas, the AMF preference is over the plant symbiont, while the abiotic conditions of soil had no effect on the composition of the fungal community.

According to Davison et al. [14], 34% of AMF species are cosmopolitan. *Acaulospora* and *Glomus* are the most represented genera in several studies including different vegetation types [5,6,12,13,15,96,113,114,115]. For example, *A. scrobiculata* is found on six continents, while in Brazil, it is recorded in all biomes [12], evidencing the global distribution of AMF, when SSU is used as a molecular marker [15].

Nevertheless, the inference of a global distribution of a certain species should be reanalyzed carefully; after all, the majority of available sequences are from rRNA (sequences of SSU or LSU), which are considered conservative when compared with other genomic sequences. Furthermore, for a phylogenetic species recognition, a multi-locus approach presents a higher resolution power for species discrimination [116]. The use of few and relatively conservative markers may give the wrong impression that everything is everywhere, which is an outdated view in mycology. There are many examples of cryptic species distribution revealed by phylogeographic, mainly for pathogenic fungi, which, unlike the AMF, count on a large number of genes and genomes sequenced [26,117]. In reality, even using the SSU sequence as a marker, a certain level of endemism is observed among *Glomeromycota* species [96,113].

The SSU marker is still one of the most available for AMF species, mainly for virtual taxa, representing more than 77% of the sequences on the MaarjAM database [11,14,57]. Notwithstanding, we ought to rethink about what we are analyzing indeed: the distribution of species or the distribution of a sequence that, due to its conservative nature, may depreciate species diversity and, in turn, overestimate the geographic distribution of some species.

The scarcity of distinct molecular markers available for species identification and differentiation in AMF [19,21,24,57,118] may be explained due to the problematic DNA extraction from spores. The quality of the collected spores in field may limit the yield of DNA, demanding the development of trap cultures and/or pure cultures [16]. However, once these cultures are acquired, there are other challenges to face, such as the maintenance of a pure culture under controlled conditions, free of contaminant fungal species [16]. As a consequence of these difficulties, most of the amplified sequences used for phylogenetic analyses are from the nrDNA multicopy region [2,3,16,18]. Furthermore, the search for new markers is quite laborious and expensive, as it demands (i) genetic material from a wide number of species, (ii) design of primers for gene amplification in all species, and (iii) time-consuming steps of cloning and sequencing.

Poor estimations of species diversity caused by morphological species recognition [20,118,119] or conservative molecular markers will certainly be overcome when variable sequences become available. To date, few species of *Glomeromycota* have their genome (*Acaulospora colombiana* and *A. morrowiae)* [120] or transcriptome (*A. morrowiae)* sequenced [121].

## 4. Phylogenetic Relationship among *Acaulospora* Species: Is There a Consensual Tree?

For the *Acaulospora* phylogeny we used an updated nrDNA sequence dataset. The NCBI Nucleotide database was consulted to obtain all available ribosomal sequences for *Acaulospora*. Using the keyword ((Acaulospora [Title]) OR Acaulospora [Organism]) AND ribosomal [Title], 1836 sequences were found in the GenBank, of different lengths.

The downloaded sequences were classified and divided by region: partial sequence SSU, ITS (=ITS1-5.8S-ITS2), partial sequence LSU and SSU-ITS-LSU (here designated as nrDNA). The separation of these sequences was carried out with reference sequences from Krüger et al. [118]. Highly incomplete and falsely annotated sequences were excluded after CD-Hit analysis and those addressed as *Acaulospora* spp., in GenBank annotation, had their species identified by accessing the original article of their description. Sequences from uncultured or not morphologically described species were not included in our phylogenetic analysis, as our objective was to compare the phylogenetic relationships of the previously described species concerning the spore wall ornamentation distribution in *Acaulospora* genus.

As the CD-HIT analysis and preliminary phylogenies using only SSU or LSU had a low resolution when distinguishing some species in monophyletic clusters (Appendix A), we carried out three additional phylogenetic analyses: one utilizing SSU-ITS-LSU (nrDNA), one simply with ITS region and another with all the available sequences of ribosomal genes, as a concatenated analysis. The access numbers of the sequences used in each phylogeny are listed on Table 5.

Alignments were carried out using Mafft online with the E-INS-I parameter. The alignment was viewed and manually edited (when required) using Mega 5.2. CD-HIT was used for all datasets (SSU, ITS, LSU and SSU-ITS-LSU fragment) to compare DNA sequences, with cut-off points of 94%, 98% and 100% similarity. These four datasets were grouped separately using CD-HIT to avoid repeating identical sequences and to exclude the incongruent sequences. The alignments used for all analyses are available in Appendix A.

Bayesian inference (BI) and maximum likelihood (ML) phylogenetic analyses were performed using CIPRES Science Gateway 3.3 [122]. GTR + G + I was used as a substitution model for both phylogenies, as well as both partitioned analyses: SSU/ITS1/5.8S/ITS2/LSU/indel and ITS1/5.8S/ITS2/indel. For ML, 1000 quick bootstrapping runs were determined using RAxML-NG 1.0.1 [123]; for the BI, a million generations were run in MrBayes 3.2 [124]. The topologies of the ML and BI trees were compared. A consensus tree was produced, showing the significant supporting values from both analysis (0.95 for BI, 70 for ML). After previewing in FigTree v. 1.4.4 [125], the phylogenetic tree was exported to Inkscape (v 0.91) for further editing, diligently honoring the scale.

The tree topology was essentially the same for our three phylogenies (nrDNA, ITS and concatenated), and for all analyses we observed that, in some cases, different sequences from the same species did not cluster together. In the nrDNA phylogeny (Figure 3), this was the case for only two species, *A. scrobiculata* (FR692352, FR692354, FR692350) and *A. lacunosa* (KP756427, KP756435, KP756584), while in the ITS phylogeny (Figure 4), this occurred for three species, *A. delicata* (JF439093, JF439203), *A. lacunosa* (KP756427, KP756435, KP756584), and *A. scrobiculata* (FR692352, FR692354, FR692350), and in the concatenated analysis, three species had their sequences clustered with others: *A. lacunosa* (KP756427, KP756435, KP756584), *A. longula* (AM040291, AM040292, AJ510228), *A. scrobiculata* (FR692352, FR692354, FR692350). Aside from this problem of “polyphyletic” species, the concatenated and ITS (Figure 4 and Figure 5) analysis also presented polytomic branches between some sister species.

Therefore, the nrDNA dataset (Figure 3) provided the best phylogeny as, in this tree, the clades presented the highest number of bootstrap supports and the lowest number of polytomic branches, as well as fewest polyphyletic species.

The polytomy observed for *A. lacunosa* and *A. scrobiculata* in all analyses (Figure 3, Figure 4 and Figure 5) suggests that these sequences are from different species with similar morphology, or perhaps, in these lineages, the nrDNA region has some convergent sites (homoplasy). A possible explanation for this is that *A. lacunosa* and *A. scrobiculata* form two complexes of crypt species, with very similar morphological characteristics.

Aside from *A. lacunosa* and *A. scrobiculata*, other polytomic incongruities were observed. For example, in the ITS tree, one of the sequences of *A. delicata* was grouped with *A. rugosa*, and in the concatenated tree, *A. longula* was shown to be paraphyletic.

The resulting data suggest that, for some groups, the phylogenetic signal of the ITS is insufficient for discriminating among different species. Taken together, said incongruities reinforce the urgency of more molecular markers and a multi-locus sequencing analysis, as, to date, most phylogenies in AMF reconstruct the evolutionary history of one single gene or sequence, and not species. As pointed out by Taylor et al. [126], real phylogenetic species recognition is carried out by the concordance of genealogies from different loci.

### 4.1. Is Kuklospora a Different Genus?

One of the objectives in reconstructing the *Acaulospora* phylogeny was to reconsider the relationship between *Acaulospora* and *Kuklospora*, a genus described by Sieverding and Oehl [46], based on morphology and spore ontogeny. The authors proposed two new genera, *Kuklospora* and *Intraspora*, and the transfer of *Entrophospora colombiana* and *Entrophospora kentinensis* to *Kuklospora*, which was placed as a sister genus of *Acaulospora*. Later, Kaonongbua et al. [48] proposed the transfer of these species to *Acaulospora* according to 28S analysis, suggesting that the spore development (acaulosporoid or entrophosporoid) is not of monophyletic character.

In all phylogenies (Figure 3, Figure 4 and Figure 5), the monophyletic nature of *Acaulospora* genus was confirmed, including the species *Kuklospora colombiana,* as sister species of *A. koskei* (both are species without ornamentation) and *Kuklospora kentinensis* as sister species of *A. aspera* and *A. spinosissima*, corroborating the analysis of 28S by Kaonongbua et al. [48], Krüger et al. [127] and other authors, such as Corazon-Guivin et al. [67], Crossay et al. [70], Lin et al. [91], Lee et al. [74] and Corazon-Guivin et al. [53], when they described *A. flava*, *A. saccata* and *A. fragilissima*, *A. tsugae*, *A. koreana*, and *A. aspera*, respectively.

Additionally, in our alignment, the difference in identity percentage between *Kuklospora* and its closest *Acaulospora* species was very low (21%) compared to differences among other genera, which reinforces that *Kuklospora* does not stand as a different genus.

Once *K. colombiana* and *K. kentinensis* are transferred to *Acaulospora* and the genus *Kuklospora* becomes invalid, the third species, *K. spinosa* [95] should also be relocated to *Acaulospora*; however, this will create a problem of homonyms with the specific epithets between *A. spinosa* and *K. spinosa*. Because *A. spinosa* was the species described first, it has nominal priority. Moreover, the absence of sequences from any molecular marker for *K. spinosa* makes a robust phylogenetic analysis still pending for its accurate phylogenetic positioning.

### 4.2. The Homoplasic Nature of Acaulospora Ornamentations

The ornamentations of the spore wall, illustrated in the nrDNA phylogenetic tree (Figure 3), do not follow the evolutionary history established by the molecular markers here used. The homoplasic nature of this morphological characteristic is observed in many groups of species in our trees. For instance, *A. kentinensis* (regular depression), *A. spinossissima* (projection) and *A. aspera* (irregular depression) are sister species and do not share the ornamentation type, as well as *A. foveata* and *A. lacunosa*, which have the depression type of ornamentation (irregular and regular, respectively) and form a monophyletic group with *A. koreana* and *A. mellea*, which are species without ornamentation.

The noted observations make it clear that spore wall ornamentations, at least in the *Acaulospora* genus, ought not to be applied for phylogenetic inferences, as they are exclusively used for morphological description of species. Furthermore, these evolutionary convergences ought to have their meaning investigated to better address questions such as: (i) is it possible the same species express different phenotypes, concerning ornamentation, depending on environmental conditions? (ii) Different ornamentations evolve independently in response to the same environmental pressures (climate, soil, plant host, etc.)? Indeed, a great deal of research must be undertaken in terms of molecular and biochemical interaction of these AMF with their substrate, but the starting point for all these studies must be a robust, well-supported, and multi-locus evolutionary analysis.

## 5. Morphological Characteristics of *Acaulospora* Species Shared with Other Genera

Once *K. colombiana* and *K. kentinensis* (now *A. colombiana* and *A. kentinensis*) are included in the *Acaulospora* genus [37,48], two types of spore formation, acaulosporoid and entrophosporoid, with one or two scars, respectively, should be considered as diagnostic characteristics of the genus.

Concerning the spore walls, as mentioned above, most *Acaulospora* species present three walls, with a Melzer reaction and a “beaded” layer, with few exceptions, as is the case with *A. colliculosa*, for which the inner wall is smooth, not beaded, and shows an absence of Melzer’s reagent.

The analysis of type collection for *A. sporocarpia* demonstrates a very distinct spore wall structure, different from the spore wall organization detected in *Acaulospora* species. Spores of *A. sporocarpia* present only two walls (vs. three in *Acaulospora*) and the inner wall is hyaline to light yellow, with laminated layers. Therefore, our findings suggest that *A. sporocarpia* does not belong to the *Acaulospora* clade, as well as *A. splendida, A. gedanensis* and *Polonospora polonica* (*Acaulospora polonica*, recently transferred to *Polonospora* genus). In all species, the spores present only two walls [37,62,64,71]. Notably, in our phylogeny, the species *A. entreriana*, which also presents only two walls, was grouped as an *Acaulospora* species. Therefore, additional phylogenetic analysis of isolates from the type location species ought to be investigated to verify this hypothesis.

Species with two spore walls, such as *Archaoespora trappei*, represent a very good example of species complexes with several sequences representing distinct clades of species ranking in *Archaeospora* [128]. Said data suggest that several isolates with similar morphology to *A. trappei* may represent new species in *Glomeromycota*, but the absence of sequences from type location isolates prevents the description of new species and makes the phylogeny ambiguous.

*Acaulospora brasiliensis* was originally described as *Ambispora brasiliensis* in Brazil [95]; however, according to the analysis of a variant Scottish isolate with similar morphology, Kruger et al. [96] transferred *A. brasiliensis* to *Acaulospora* (*Diversisporales*). Phylogenetic analysis of the Brazilian fungus is not available; however, its morphology does indeed resemble *Ambispora* species, not *Acaulospora*. Additional analyses of the fungus isolated in Brazil are necessary to clarify this inconsistency between the morphologies of Brazilian and Scottish isolates.

*Acaulospora terricola* represents another problematic case that invites further scrutiny. The available morphological description in protologue does not allow conclusions regarding its membership to *Acaulospora* [92]. Unlike other *Acaulospora* species with molecular data available, *A. terricola* presents ornamentation in the inner layer of its spore wall. The closest morphological species is *A. endographis*, which, similar to other *Acaulospora* species, has three walls and beaded layers, and, as ornamentations, the inner layer of outer wall of spore wall presents dense spine projections. However, *A. terricola* presents a much more complex structure, with 10 layers in the spore wall, while *A. endographis* presents only four.

*Acaulospora walkeri*, in addition, requires elucidation regarding its spore wall structure. Originally, the fungus was described with only four layers distributed in two walls, but the pictures presented in protologue suggest three walls [94].

Taking all these exceptions into account, we can rule out any morphological characteristic, to date, as being a real apomorphy of the *Acaulospora* genus. The molecular markers appear to present a better resolution. Our analyses showed a robust resolution for the monophyly of this genus, because, as stated previously, a few sparse cases in our analyses remained polyphyletic.

Poor-quality or insufficiently annotated sequences in GenBank may present another problem that would corroborate the occurrence of polyphyletic species in phylogenies. By applying a similarity cut-off of 98% for all downloaded sequences, we concluded that some may present entirely different nucleotides for the SSU and LSU regions. Indeed, in our first phylogenies, these sequences did not group even within the *Acaulospora* genus (data not shown). We call them “contaminating sequences”, as they completely altered our alignment. The access number of these sequences in GenBank are as follows: Z14006.1, NG_062371.1, HE610427.1, Y17633.2, AJ306439.1, FJ009670.1, NG_062381.1, Z14005.1, corresponding to the species *A. colombiana, A. cavernata, A. lacunosa, A. laevis, A. longula, A. mellea, A. spinosa* and *A. rugosa*, respectively. It is possible that they are sequences from other genera of *Glomeromycota* that have mistakenly been identified as *Acaulospora*.

Therefore, the aforementioned discrepancies warrant further investigation when studying *Acaulospora* species. Despite the high phylogenetic support of the genus, the resolution of some species relationships is still a challenge, perhaps due to factors such as cryptic speciation, phenotypic plasticity and genetic homoplasic polymorphism. To solve this enigma, more genes, such as *RPB1*, *Alfa tubulin* and *Beta tubulin*, should be sequenced and used in a multi-locus approach.

## 6. Conclusions

*Acaulospora* species are found in 61 countries around the world, foremost in temperate and tropical zones with the greatest record of diversity of this genus. Brazil is the country with the highest recorded diversity (43/60), showcasing a great potential for describing new species of *Acaulospora* and other AMF.

For the *Acaulospora* genus, the nrDNA tree containing SSU, ITS1, 5.8S, ITS2 and LSU gene sequences proved to be superior as compared to the concatenated analysis and the ITS-only tree, due to the lowest number of polyphyletic species found in this phylogeny. This polyphylism may be due to the occurrence of morphospecies or the poorly annotated sequences in the databases.

The phylogenetic trees revealed the homoplasic nature of the spore wall ornamentation in *Acaulospora* genus, indicating that it ought not to be used as a phylogenetic marker. We supported the inclusion of *Kuklospora* species as belonging to *Acaulospora*, using a robust phylogenetic analysis of the nrDNA region. Further molecular analysis is required to clarify the position and phylogenetic relationship of *K. spinosa*.

Lastly, we emphasize the importance of genome sequencing of more AMF species, as well as the sequencing of other markers for robust multi-locus phylogenies and, therefore, a better understanding of the evolution of these fungi as a starting point for clarifying the possible ecological meanings of morphological convergences, such as the ornamentation found among the species from the *Acaulospora* genus studied here.

## Figures and Tables

**Figure 1 jof-08-00892-f001:**
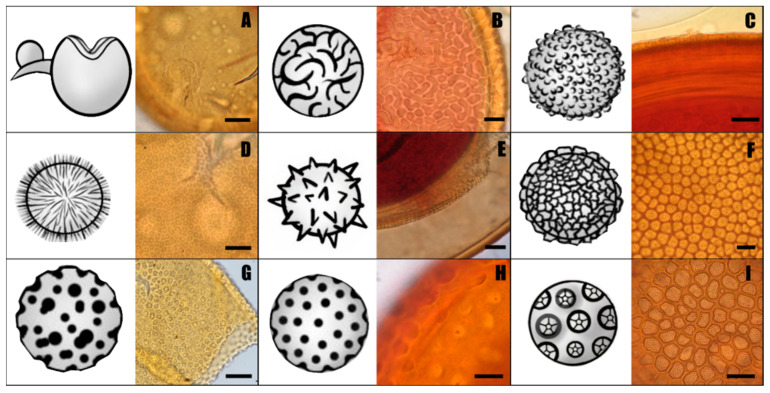
Spore wall ornamentations in *Acaulospora* species. (**A**) Smooth outer layer in *A. morrowiae*. (**B**) Cerebriform ornamentation in *A. rhemii*. (**C**) Spinose ornamentation in *A. tuberculata*. (**D**) Double ornamentation in *A. elegans*. (**E**) Spinose projection of the inner layers in *A. endographis*. (**F**) Denticulate (teeth) ornamentation in *A. denticulata*. (**G**) Dense irregular pits in *A. herrerae*. (**H**) Pits in the *A. foveata*. (**I**) Double ornamentation in *A. bireticulata*. Scale bars in (**A**,**C**,**D**,**G**–**I**) represent 10 µm. Scale bars in (**B**,**E**,**F**) represent 5 µm.

**Figure 2 jof-08-00892-f002:**
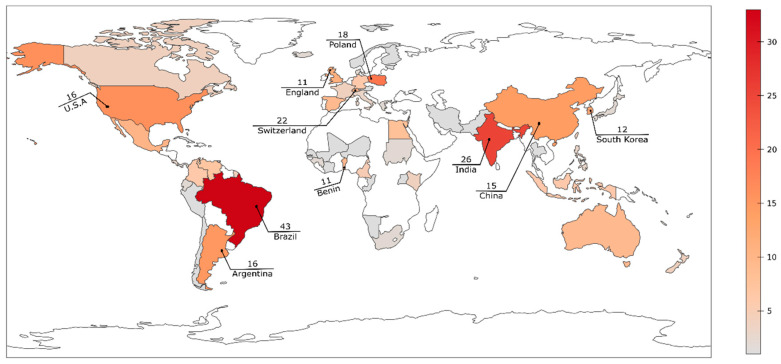
Global distribution of *Acaulospora* species. The numbers indicate the total *Acaulospora* species in the ten countries with the highest occurrence rates.

**Figure 3 jof-08-00892-f003:**
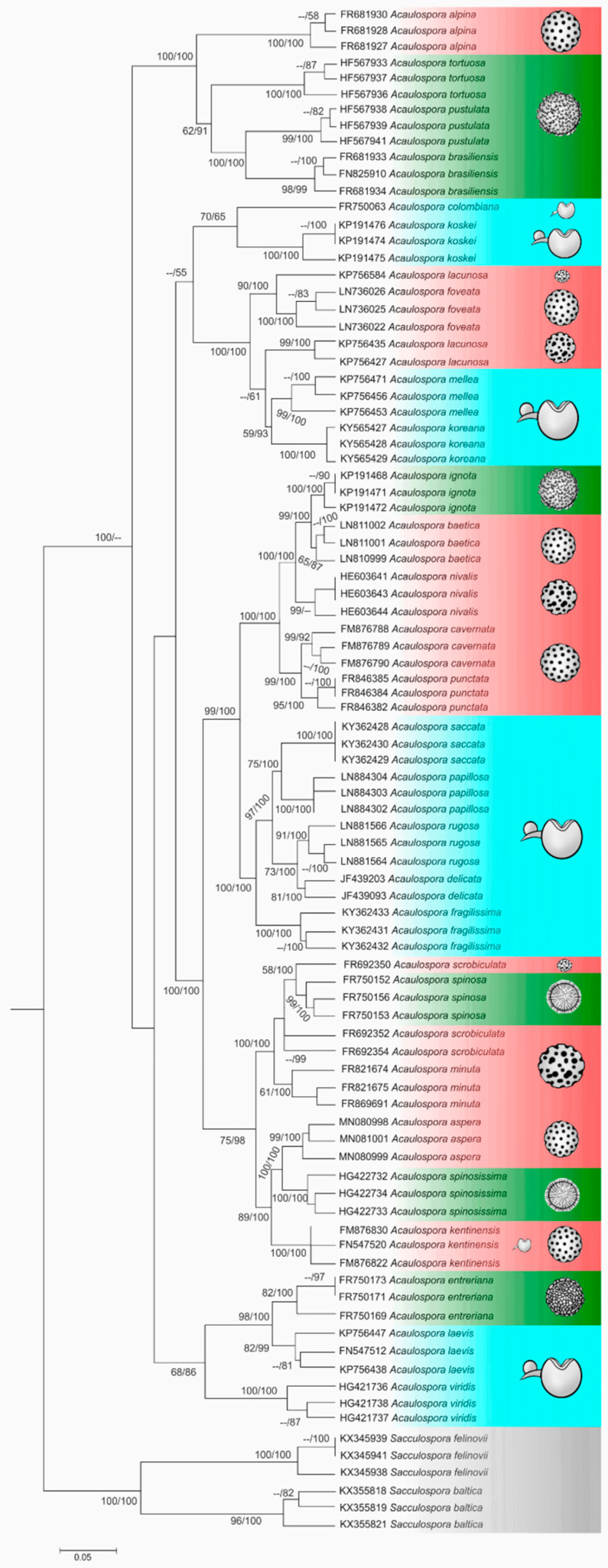
Phylogenetic tree of *Acaulospora* species using nrDNA sequences (partial SSU, ITS1, 5.8S, ITS2 and partial LSU). The alignment was performed and contained twenty-nine *Acaulospora* species and two *Sacculospora* species as outgroups. The support values are Bayesian inference (BI) and maximum likelihood (ML), respectively, with values equal to or higher than 0.95 for BI and 70 for ML considered significant. *Sacculospora baltica* and *Sacculospora felinovii* were included as outgroups. Blue—smooth spores; Green—projection-shaped ornamentation; Red—depression-shaped ornamentation; Grey—outgroup.

**Figure 4 jof-08-00892-f004:**
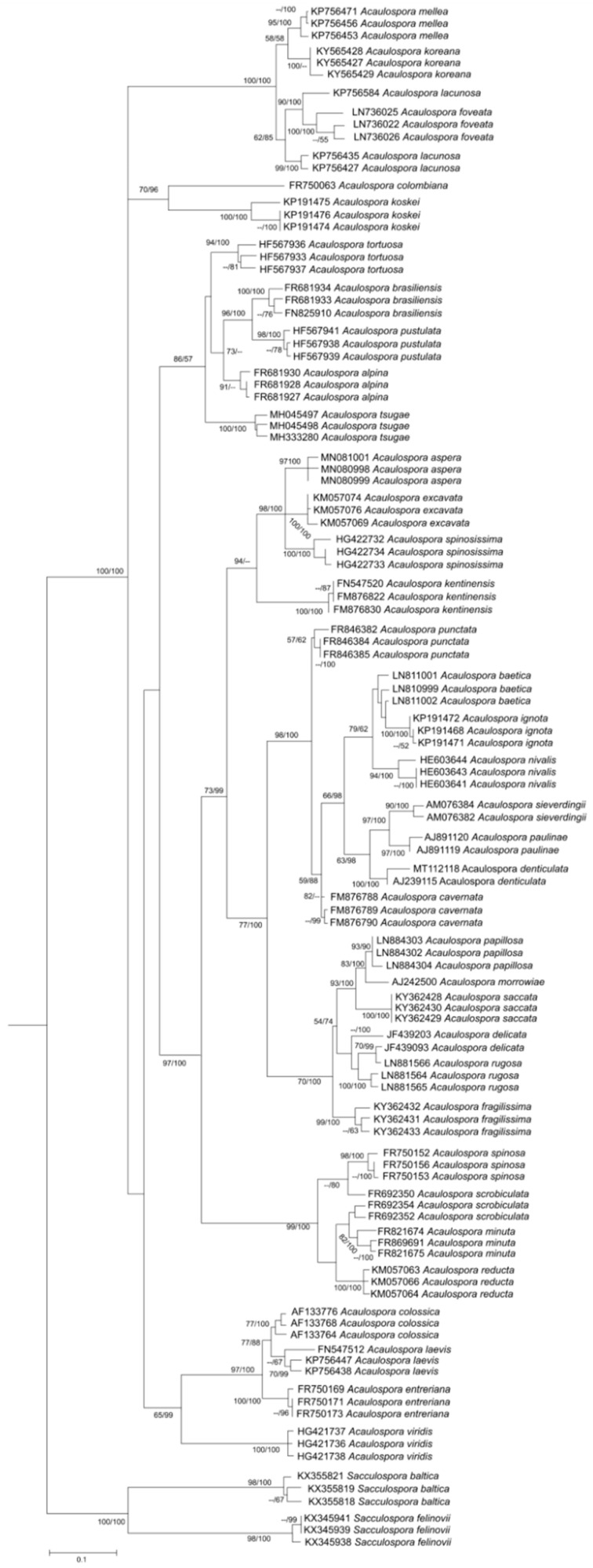
Phylogenetic tree using complete ITS1-5.8S-ITS2 sequences from thirty-five *Acaulospora* species and two *Sacculospora* species as outgroups. The support values are Bayesian inference (BI) and maximum likelihood (ML), respectively, with values ≥0.95 for BI and ≥70 for ML considered significant.

**Figure 5 jof-08-00892-f005:**
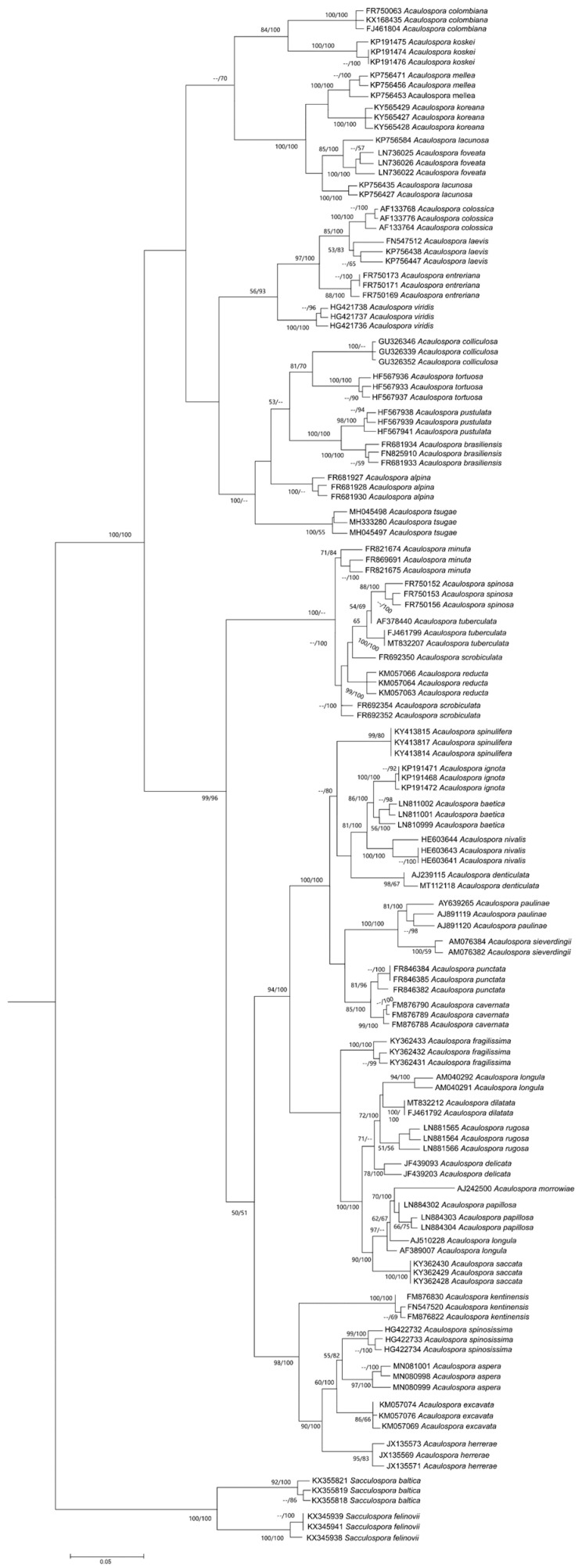
Phylogenetic tree using all available nrDNA sequences (partial SSU, ITS, partial LSU, SSU-ITS-LSU) in a concatenated analysis. Forty-three *Acaulospora* species were included, along with *Sacculospora baltica* and *Sacculospora felinovii,* as outgroups. The support values are Bayesian inference (BI) and maximum likelihood (ML), respectively, with values ≥0.95 for BI and ≥70 for ML considered significant.

**Table 1 jof-08-00892-t001:** Morphologically analyzed species and their reference numbers.

Species	Reference Number
*Acaulospora alpina*	UFRN-Fungos 3408
*Acaulospora ignota*	UFRN-Fungos 3409
*Acaulospora denticulata*	UFRN-Fungos 3410
*Acaulospora excavata*	UFRN-Fungos 3411
*Acaulospora elegans*	UFRN-Fungos 3412
*Acaulospora foveata*	UFRN-Fungos 3413
*Acaulospora herrerae*	UFRN-Fungos 3414
*Acaulospora laevis*	UFRN-Fungos 3415
*Acaulospora lacunosa*	UFRN-Fungos 3416
*Acaulospora mellea*	UFRN-Fungos 3417
*Acaulospora morrowiae*	UFRN-Fungos 3418
*Acaulospora reducta*	UFRN-Fungos 3419
*Acaulospora rugosa*	UFRN-Fungos 3420
*Acaulospora scrobiculata*	UFRN-Fungos 3421
*Acaulospora spinosa*	UFRN-Fungos 3422
*Acaulospora spinosissima*	UFRN-Fungos 2965
*Acaulospora sporocarpia*	OSC, Oregon, 47836 ^1^
*Acaulospora tuberculata*	UFRN-Fungos 3423

^1^ Isotype.

**Table 2 jof-08-00892-t002:** Description and classification of ornaments of the *Acaulospora* species.

Species	Ornate	Type	Shape	Original Description
*Acaulospora alpina*	Yes	Depressions	Pits	regular conical pits
*Acaulospora aspera*	Yes	Depressions	Pits	irregular depressions
*Acaulospora baetica*	Yes	Depressions	Pits	pitted
*Acaulospora bireticulata*	Yes	Double depressions	Depressed with central stratum	polygonal reticulum depressed central stratum; ridges occasionally branched forming irregular isolated projections
*Acaulospora brasiliensis*	Yes	Projections	Pustules	convex pustules irregular shape and size
*Acaulospora capsicula*	No	---	---	---
*Acaulospora cavernata*	Yes	Depressions	Pits	evenly pitted deep depressions separated by ridges
*Acaulospora colliculosa*	Yes	Projections	Protuberances	protuberances
*Acaulospora colombiana*	No	---	---	---
*Acaulospora colossica*	No	---	---	---
*Acaulospora delicata*	No	---	---	---
*Acaulospora denticulata*	Yes	Projections	Teeth	tooth-shaped projections circular or oblong
*Acaulospora dilatata*	No	---	---	---
*Acaulospora endographis*	Yes	Projections	Spines	irregular spines
*Acaulospora elegans*	Yes	Projections	Spines	ornamented with crowded densely organized spines
*Acaulospora entreriana*	Yes	Projections	Teeth	ornamented with teeth
*Acaulospora excavata*	Yes	Depressions	Pits	concave round pits
*Acaulospora fanjing*	No	---	---	---
*Acaulospora flava*	No	---	---	---
*Acaulospora flavopapillosa*	Yes	Projections	Papillae	fine papillae
*Acaulospora foveata*	Yes	Depressions	Pits	round to oblong and concave depressions
*Acaulospora fragilissima*	No	---	---	---
*Acaulospora gedanensis*	No	---	---	---
*Acaulospora herrerae*	Yes	Depressions	Pits	rounded to elliptical pits some pits vermiform or regulate
*Acaulospora ignota*	Yes	Projections	Excrescences	granular excrescences
*Acaulospora kentinensis*	Yes	Depressions	Pits	pits circular to subcircular deep when observed in cross view
*Acaulospora koreana*	No	---	---	---
*Acaulospora koskei*	No	---	---	---
*Acaulospora lacunosa*	Yes	Depressions	Pits	irregularly distributed irregular saucer-shaped pits
*Acaulospora laevis*	No	---	---	---
*Acaulospora longula*	No	---	---	---
*Acaulospora mellea*	No	---	---	---
*Acaulospora minuta*	Yes	Depressions	Pits	minute pit-like depressions
*Acaulospora morrowiae*	No	---	---	---
*Acaulospora nivalis*	Yes	Depressions	Pits	irregular pits
*Acaulospora papillosa*	No	---	---	---
*Acaulospora paulinae*	Yes	Depressions	Pits	concave round pits of widest diameter
*Acaulospora punctata*	Yes	Depressions	Pits	regular round pits
*Acaulospora pustulata*	Yes	Projections	Blister	pustulate projections
*Acaulospora reducta*	Yes	Depressions	Pits	irregularly-shaped small pits sometimes dumbbell-shaped pits
*Acaulospora rehmii*	Yes	Projections	Cerebriform	cerebriform folds
*Acaulospora rugosa*	No	---	---	---
*Acaulospora saccata*	No	---	---	---
*Acaulospora scrobiculata*	Yes	Depressions	Pits	ornamented with evenly distributed pits, circular ellipsoidal oblong triangular Y-shaped to irregular
*Acaulospora sieverdingii*	Yes	Depressions	Pits	irregular pits
*Acaulospora soloidea*	Yes	Projections	Bristle	ornamented with numerous acellular fibrillose hairy outgrowths forming a pile or thick coat
*Acaulospora spinosa*	Yes	Projections	Spines	densely organized spines
*Acaulospora spinosissima*	Yes	Projections	Spines	short spiny projections
*Acaulospora spinulifera*	Yes	Projections	Spines	fine spines
*Acaulospora splendida*	No	---	---	---
*Acaulospora sporocarpia*	No	---	---	---
*Acaulospora taiwania*	Yes	Depressions	Pits	side pits ridges form mesh
*Acaulospora tsugae*	No	---	---	---
*Acaulospora terricola*	Yes	Depressions	Pits	minutely pitted
*Acaulospora tortuosa*	Yes	Projections	Excrescences	tortuous hyphae-like structures on the surface
*Acaulospora thomii*	No	---	---	---
*Acaulospora tuberculata*	Yes	Projections	Spines/Tubercles	ornamented with evenly spines or tubercles
*Acaulospora viridis*	No	---	---	---
*Acaulospora verna*	Yes	Depressions	Pits	ornamented with evenly distributed pits, circular to subcircular frequently ellipsoidal to oblong sometimes irregular
*Acaulospora walkeri*	Yes	Projections	Excrescences	finely ornamented
*Kuklospora spinosa*	Yes	Projections	Spines	fine spines

**Table 3 jof-08-00892-t003:** Overview of the species transferred from *Acaulospora* to other genera.

Original Name	Description	Current Name	Description	Authors
*≡Acaulospora appendicula*	1984	*Ambispora appendicula*	2008	(Spain, Sieverd. & N.C. Schenck) C. Walker
*≡Acaulospora gerdemannii*	1979	*Ambispora jimgerdemannii*	2008	(Spain, Oehl & Sieverd.) C. Walker
*≡Acaulospora myriocarpa*	1986	*Archaeospora myriocarpa*	2011	(Spain, Sieverd. & N.C. Schenck) Oehl, G.A. Silva, B.T. Goto & Sieverd.
*≡Acaulospora nicolsonii*	1984	*Ambispora nicolsonii*	2012	(C. Walker, L.E. Reed & F.E. Sanders) Oehl, G.A. Silva, B.T. Goto & Sieverd.
*≡Acaulospora trappei*	1976	*Archaeospora trappei*	2001	(R.N. Ames & Linderman) J.B. Morton & D. Redecker
*≡Acaulospora undulata*	1988	*Archaeospora undulata*	2011	(Sieverd.) Sieverd., G.A. Silva, B.T. Goto & Oehl
*≡Ambispora brasiliensis*	2008	*Acaulospora brasiliensis*	2011	(B.T. Goto, L.C. Maia & Oehl) C. Walker, Krüger & Schüßler
*≡Acaulospora polonica*	1988	*Polonospora polonica*	2021	(Błaszk.) Błaszk., Niezgoda, B.T. Goto & Magurno

**Table 4 jof-08-00892-t004:** Global distribution of *Acaulospora* species and sequences available in online databases. (Available in: NCBI, EMBL, GBIF, BLOYD SYSTEMS, MaarJAM, MYCOBANK).

N°	AMF Species	Occurrence	Sequence	Reference
1	*Acaulospora alpina* Oehl, Sýkorová & Sieverd.	England, Switzerland, India, Brazil	SSU-ITS-LSU	[52]
2	*Acaulospora aspera* Corazon-Guivin, Oehl & G.A. Silva	Peru	SSU-ITS-LSU	[53]
3	*Acaulospora baetica* Palenz., Oehl, Azcón-Aguilar & G.A. Silva.	Spain, Brazil	SSU-ITS-LSU	[54]
4	*Acaulospora bireticulata* F.M. Rothwell & Trappe	Brazil, Argentina, England, South Korea, Poland, Egypt, India, United States, China, Italy	---	[37,55]
5	*Acaulospora brasiliensis* (B.T. Goto, L.C. Maia & Oehl) C. Walker, Krüger & Schüßler	Brazil, Scotland, Argentina, South Korea	SSU-ITS-LSU	[56,57]
6	*Acaulospora capsicula* Błaszk.	Australia, England, Poland, China, Brazil, Egypt, India, United States, Switzerland	---	[58]
7	*Acaulospora cavernata* Błaszk.	Poland, Brazil, Benin, China, Switzerland	SSU-ITS-LSU	[59]
8	*Acaulospora colliculosa* Kaonongbua, J.B. Morton & Bever	United States, England	LSU	[48]
9	*Acaulospora colombiana* (Spain & N.C. Schenck) Kaonongbua, J.B. Morton & Bever	Colombia, Brazil, India, Philippines, Benin, Germany, Switzerland	SSU-ITS-LSU; ORF1 gene *cox1*; *Beta Tubulin*; gene *CHS*	[48]
10	*Acaulospora colossica* P.A. Schultz, Bever & J.B. Morton	United States, Brazil	SSU ^1^-ITS	[60]
11	*Acaulospora delicata* C. Walker, C.M. Pfeiffer & Bloss	Australia, Argentina, Brazil, China, United States, Spain, Philippines, Indonesia, Mexico, England, Senegal, Venezuela, Poland, India, Iceland	SSU-ITS-LSU	[61]
12	*Acaulospora denticulata* Sieverd. & S. Toro	Argentina, Colombia, Mexico, Brazil, South Korea, South Africa, China, India, Canada, Kenya	ITS	[62]
13	*Acaulospora dilatata* J.B. Morton	Argentina, United States, New Zealand, South Atlantic, Venezuela, Brazil, India	SSU; LSU; *Beta tubulin*	[63]
14	*Acaulospora endographis* B.T. Goto	Brazil	---	[49]
15	*Acaulospora elegans* Trappe & Gerd.	United States, Indonesia, Poland, Brazil, South Korea, Argentina, India, South Africa, Switzerland	LSU	[27]
16	*Acaulospora entreriana* M.S. Velázquez & Cabello	Argentina, Brazil	SSU-ITS-LSU	[64]
17	*Acaulospora excavata* Ingleby & C. Walker	Australia, Argentina, Ivory Coast, Spain, New Zealand, Trinidad and Tobago, Brazil	ITS	[65]
18	*Acaulospora fanjing* R.J. He, L.M. Yao & L. Jiang	China	SSU-ITS-LSU	[66]
19	*Acaulospora flava* Corazon-Guivin, G.A. Silva & Oehl	Peru	SSU-ITS-LSU	[67]
20	*Acaulospora flavopapillosa* Corazon-Guivin, G.A. Silva & Oehl	Peru	SSU-ITS-LSU	[68]
21	*Acaulospora foveata* Trappe & Janos	Australia, Brazil, Costa Rica, Mexico, England, Congo, Poland, Panama, Indonesia, South Korea, Argentina, India	SSU-ITS-LSU	[69]
22	*Acaulospora fragilissima* D. Redecker, Crossay & Cilia	New Caledonia, Peru	SSU-ITS-LSU; *Beta tubulin*; ORF1 gene *cox1*	[70]
23	*Acaulospora gedanensis* Błaszk.	Poland, Brazil, Austria, Switzerland	---	[71]
24	*Acaulospora herrerae* Furrazola, B.T. Goto, G.A. Silva, Sieverd. & Oehl	Brazil, Cuba, Peru	LSU	[72]
25	*Acaulospora ignota* Błaszk., Góralska, Chwat & B.T. Goto	Brazil	SSU-ITS-LSU	[73]
26	*Acaulospora kentinensis* (Wu & Liu) Oehl & Sieverd.	Taiwan, Brazil, Benin	SSU-ITS-LSU; ORF1 gene *cox1*; *Beta Tubulin*	[48]
27	*Acaulospora koreana* E. H. Lee, S. H. Park & A. H	North Korea	SSU-ITS-LSU	[74]
28	*Acaulospora koskei* Błaszk.	Australia, Spain, Poland, Brazil, Estonia, Egypt, India, Cameroon, England	SSU-ITS-LSU; *Beta tubulin*	[75]
29	*Acaulospora lacunosa* J.B. Morton	Argentina, United States, New Zealand, Venezuela, Brazil, Poland, Denmark, India, Kenya, China	SSU-ITS-LSU; *Beta tubulin*; *HSP60* gene	[63]
30	*Acaulospora laevis* Gerd. & Trappe	Australia, Argentina, China, Ecuador, United States, New Zealand, England, Brazil, South Korea, Egypt, India, Mexico, Iceland, France, Germany, Greece, Benin, Switzerland, Israel, South Africa	SSU-ITS-LSU; *Beta tubulin*; *RPB1*; mRNA gene *Ste12*; *Alfa tubulin;ef1 Alpha*	[27]
31	*Acaulospora longula* Spain & N.C. Schenck	Australia, Germany, Brazil, Colombia, Spain, Philippines, Japan, Indonesia, England, Uganda, Venezuela, South Korea, India, Switzerland	SSU; LSU; *RPB1*	[76]
32	*Acaulospora mellea* Spain & N.C. Schenck	Germany, Argentina, Brazil, Colombia, China, United States, Mexico, Poland, India, Canada, Denmark, Benin, Cameroon, Switzerland	SSU-ITS-LSU; *Beta tubulin*; ORF1 gene *cox1*	[76]
33	*Acaulospora minuta* Oehl, Tchabi, Hount., Palenz., I.C. Sánchez & G.A. Silva	Benin, Brazil	SSU-ITS-LSU	[77]
34	*Acaulospora morrowiae* Spain &N.C. Schenck	Colombia, United States, Mexico, Brazil, Poland, South Korea, Namibia, India, Niger, Mali, France, Germany, Switzerland, Benin, Cameroon, China	SSU; LSU; SSU-ITS; ITS-LSU ^1^; *Beta tubulin*; *glomalin* cDNA; *HSP60* gene	[76]
35	*Acaulospora nivalis* Oehl, Palenz., I.C. Sánchez, G.A. Silva & Sieverd.	Switzerland, Brazil	SSU-ITS-LSU	[78]
36	*Acaulospora papillosa* C.M.R. Pereira & Oehl	Brazil, Norway	SSU-ITS-LSU	[79]
37	*Acaulospora paulinae* Błaszk.	Brazil, Poland, United States, India, Germany, France, Switzerland, Israel	ITS2-LSU; SSU-ITS1; SSU-ITS	[80]
38	*Acaulospora punctata* Oehl, Palenz., I.C. Sánchez, G.A. Silva, C. Castillo & Sieverd.	Brazil, Switzerland, Chile	SSU-ITS-LSU	[81]
39	*Acaulospora pustulata* Palenz., Oehl, Azcon-Aguilar & G.A. Silva	Spain, Switzerland	SSU-ITS-LSU	[82]
40	*Acaulospora reducta* Oehl, B.T. Goto & C.M.R. Pereira	Brazil	SSU ^1^-ITS-LSU ^1^	[83]
41	*Acaulospora rehmii* Sieverd. & S. Toro	Brazil, Mexico, South Korea, Argentina, Poland, Colombia, Taiwan, Egypt, India, China	---	[62]
42	*Acaulospora rugosa* J.B. Morton	Brazil, United States, South Korea, Poland, India, Iran	SSU-ITS-LSU	[63]
43	*Acaulospora saccata* D. Redecker, Crossay & Cilia	New Caledonia	SSU-ITS-LSU	[70]
44	*Acaulospora scrobiculata* Trappe	Brazil, England, Mexico, Australia, Indonesia, Trinidad and Tobago, Spain, Argentina, Venezuela, United States, Thailand, Japan, Philippines, Poland, Canada, Israel, Greece, Italy, Cameroon, China, Taiwan, South Korea, India, Benin, Finland, Kenya, Switzerland, Nepal	SSU-ITS-LSU; *Beta tubulin*; gene *CHS*	[84]
45	*Acaulospora sieverdingii* Oehl, Sýkorová & Błaszk.	Brazil, Germany, Poland, Italy, Northern Guinea, Southern Guinea, Benin, France, Sudan, Switzerland	ITS	[47]
46	*Acaulospora**soloidea* Vaingankar & B.F. Rodrigues	India, Egypt	---	[85]
47	*Acaulospora spinosa* C. Walker & Trappe	United States, Mexico, Argentina, Brazil, South Korea, Ecuador, China, India, Canada, Venezuela, Colombia, Benin, Cameroon, Kenya, South Africa, Switzerland, Nepal	SSU-ITS-LSU	[86]
48	*Acaulospora spinosissima* Oehl, Palenz., Sánchez-Castro, Tchabi, Hount. & G. A. Silva	Sudan, South Africa, Northern Guinea, Southern Guinea, Switzerland, Benin, Brazil	SSU-ITS-LSU	[53]
49	*Acaulospora spinulifera* Oehl, V.M. Santos, J.S. Pontes & G.A. Silva	Brazil	LSU	[87]
50	*Acaulospora splendida* Sieverd., Chaverri & I. Rojas	Costa Rica, India, Mexico, Egypt, Brazil	---	[88]
51	*Acaulospora sporocarpia* S.M. Berch	England, Switzerland, United States, Pakistan, Egypt, India, Brazil	---	[89]
52	*Acaulospora taiwania* H.T. Hu	Taiwan	---	[90]
53	*Acaulospora tsugae* T.C.Lin & Oehl	Taiwan, Switzerland	SSU ^1^-ITS-LSU ^1^	[91]
54	*Acaulospora terricola* Swarupa, Kunwar & Manohar	India	---	[92]
55	*Acaulospora tortuosa* Palenz., Oehl, Azcon-Aguilar & G.A.Silva	Spain, Switzerland	SSU-ITS-LSU	[82]
56	*Acaulospora thomii* Błaszk.	Poland, Egypt, India, Switzerland	---	[71]
57	*Acaulospora tuberculata* Janos & Trappe	Brazil, Venezuela, Argentina, Costa Rica, Panama, China, Egypt, India	LSU	[69]
58	*Acaulospora viridis* Palenz., Oehl, Azcón-Aguilar & G.A.Silva	Spain	SSU-ITS-LSU	[93]
59	*Acaulospora verna* Błaszk.	Poland	---	[37]
60	*Acaulospora walker* Kramad. & Hedger	Australia, Indonesia, Brazil	---	[94]
	*Kuklospora spinosa* B.P. Cai, Jun Y. Chen, Q.X. Zhang & L.D. Guo	China	---	[95]

^1^ Short sequences with less than 50 bp.

**Table 5 jof-08-00892-t005:** Sequences used in phylogenetic analyses.

Access Number Sequence	Concatenated Analysis	SSU-ITS-LSU	ITS1-5.8S-ITS2
KX355819_*Sacculospora_baltica*	x	x	x
KX355818_*Sacculospora_baltica*	x	x	x
KX355821_*Sacculospora_baltica*	x	x	x
KX345938_*Sacculospora_felinovii*	x	x	x
KX345939_*Sacculospora_felinovii*	x	x	x
KX345941_*Sacculospora_felinovii*	x	x	x
FR681927_*Acaulospora_alpina*	x	x	x
FR681928_*Acaulospora_alpina*	x	x	x
FR681930_*Acaulospora_alpina*	x	x	x
MN080998_*Acaulospora_aspera*	x	x	x
MN081001_*Acaulospora_aspera*	x	x	x
MN080999_*Acaulospora_aspera*	x	x	x
LN810999_*Acaulospora_baetica*	x	x	x
LN811001_*Acaulospora_baetica*	x	x	x
LN811002_*Acaulospora_baetica*	x	x	x
FN825910_*Acaulospora_brasiliensis*	x	x	x
FR681934_*Acaulospora_brasiliensis*	x	x	x
FR681933_*Acaulospora_brasiliensis*	x	x	x
FM876789_*Acaulospora_cavernata*	x	x	x
FM876790_*Acaulospora_cavernata*	x	x	x
FM876788_*Acaulospora_cavernata*	x	x	x
AF133764_*Acaulospora_colossica*	x	x	x
AF133768_*Acaulospora_colossica*	x	x	x
AF133776_*Acaulospora_colossica*	x	x	x
GU326339_*Acaulospora_colliculosa*	x		
GU326346_*Acaulospora_colliculosa*	x		
GU326352_*Acaulospora_colliculosa*	x		
FR750063_*Acaulospora_colombiana*	x	x	x
FJ461804_*Acaulospora_colombiana*	x		
KX168435_*Acaulospora_colombiana*	x		
JF439093_*Acaulospora_delicata*	x	x	x
JF439203_*Acaulospora_delicata*	x	x	x
MT832212_*Acaulospora_dilatata*	x		
FJ461792_*Acaulospora_dilatata*	x		
AJ239115_*Acaulospora_denticulata*	x		x
MT112118_*Acaulospora_denticulata*	x		x
FR750173_*Acaulospora_entreriana*	x	x	x
FR750171_*Acaulospora_entreriana*	x	x	x
FR750169_*Acaulospora_entreriana*	x	x	x
KM057069_*Acaulospora_excavata*	x		x
KM057074_*Acaulospora_excavata*	x		x
KM057076_*Acaulospora_excavata*	x		x
KY362433_*Acaulospora_fragilissima*	x	x	x
KY362432_*Acaulospora_fragilissima*	x	x	x
KY362431_*Acaulospora_fragilissima*	x	x	x
LN736022_*Acaulospora_foveata*	x	x	x
LN736026_*Acaulospora_foveata*	x	x	x
LN736025_*Acaulospora_foveata*	x		
JX135571_*Acaulospora_herrerae*	x		
JX135569_*Acaulospora_herrerae*	x		
JX135573_*Acaulospora_herrerae*	x		
KP191468_*Acaulospora_ignota*	x	x	x
KP191471_*Acaulospora_ignota*	x	x	x
KP191472_*Acaulospora_ignota*	x	x	x
FM876830_*Acaulospora_kentinensis*	x	x	x
FM876822_*Acaulospora_kentinensis*	x	x	x
FN547520_*Acaulospora_kentinensis*	x	x	x
KP191475_*Acaulospora_koskei*	x	x	x
KP191474_*Acaulospora_koskei*	x	x	x
KP191476_*Acaulospora_koskei*	x	x	x
KY565428_*Acaulospora_koreana*	x	x	x
KY565427_*Acaulospora_koreana*	x	x	x
KY565429_*Acaulospora_koreana*	x	x	x
KP756438_*Acaulospora_laevis*	x	x	x
KP756447_*Acaulospora_laevis*	x	x	x
FN547512_*Acaulospora_laevis*	x	x	x
KP756427_*Acaulospora_lacunosa*	x	x	x
KP756435_*Acaulospora_lacunosa*	x	x	x
KP756584_*Acaulospora_lacunosa*	x	x	x
AM040291_*Acaulospora_longula*	x		
AM040292_*Acaulospora_longula*	x		
AJ510228_*Acaulospora_longula*	x		
AF389007_*Acaulospora_longula*	x		
KP756453_*Acaulospora_mellea*	x	x	x
KP756456_*Acaulospora_mellea*	x	x	x
KP756471_*Acaulospora_mellea*	x	x	x
FR869691_*Acaulospora_minuta*	x	x	x
FR821675_*Acaulospora_minuta*	x	x	x
FR821674_*Acaulospora_minuta*	x	x	x
AJ242500_*Acaulospora_morrowiae*	x		x
HE603641_*Acaulospora_nivalis*	x	x	x
HE603643_*Acaulospora_nivalis*	x	x	x
HE603644_*Acaulospora_nivalis*	x	x	x
AJ891120_*Acaulospora_paulinae*	x		x
AJ891119_*Acaulospora_paulinae*	x		x
AY639265_*Acaulospora_paulinae*	x		x
LN884304_*Acaulospora_papillosa*	x	x	x
LN884303_*Acaulospora_papillosa*	x	x	x
LN884302_*Acaulospora_papillosa*	x	x	x
FR846382_*Acaulospora_punctata*	x	x	x
FR846384_*Acaulospora_punctata*	x	x	x
FR846385_*Acaulospora_punctata*	x	x	x
HF567941_*Acaulospora_pustulata*	x	x	x
HF567939_*Acaulospora_pustulata*	x	x	x
HF567938_*Acaulospora_pustulata*	x	x	x
KM057064_*Acaulospora_reducta*	x		x
KM057066_*Acaulospora_reducta*	x		x
KM057063_*Acaulospora_reducta*	x		x
LN881566_*Acaulospora_rugosa*	x	x	x
LN881565_*Acaulospora_rugosa*	x	x	x
LN881564_*Acaulospora_rugosa*	x	x	x
KY362428_*Acaulospora_saccata*	x	x	x
KY362430_*Acaulospora_saccata*	x	x	x
KY362429_*Acaulospora_saccata*	x	x	x
FR692352_*Acaulospora_scrobiculata*	x	x	x
FR692354_*Acaulospora_scrobiculata*	x	x	x
FR692350_*Acaulospora_scrobiculata*	x	x	x
AM076384_*Acaulospora_sieverdingii*	x		x
AM076382_*Acaulospora_sieverdingii*	x		x
FR750153_*Acaulospora_spinosa*	x	x	x
FR750156_*Acaulospora_spinosa*	x	x	x
FR750152_*Acaulospora_spinosa*	x	x	x
HG422734_*Acaulospora_spinosissima*	x	x	x
HG422733_*Acaulospora_spinosissima*	x	x	x
HG422732_*Acaulospora_spinosissima*	x	x	x
KY413817_*Acaulospora_spinulifera*	x		
KY413815_*Acaulospora_spinulifera*	x		
KY413814_*Acaulospora_spinulifera*	x		
HF567933_*Acaulospora_tortuosa*	x	x	x
HF567937_*Acaulospora_tortuosa*	x	x	x
HF567936_*Acaulospora_tortuosa*	x	x	x
MH045497_*Acaulospora_tsugae*	x		x
MH045498_*Acaulospora_tsugae*	x		x
MH333280_*Acaulospora_tsugae*	x		x
AF378440_*Acaulospora_tuberculata*	x		
FJ461799_*Acaulospora_tuberculata*	x	x	x
MT832207_*Acaulospora_tuberculata*	x	x	x
HG421736_*Acaulospora_viridis*	x	x	x
HG421738_*Acaulospora_viridis*	x	x	x
HG421737_*Acaulospora_viridis*	x	x	x

## Data Availability

The databases consulted were NCBI (https://www.ncbi.nlm.nih.gov/ accessed on 20 October 2020); EMBL (https://www.ebi.ac.uk/ accessed on 20 October 2020); GBIF (https://www.gbif.org/search?q=Acaulospora accessed on 20 October 2020); BLOYD SYSTEMS (http://boldsystems.org/index.php accessed on 20 October 2020); MaarJAM (https://maarjam.botany.ut.ee accessed on 20 October 2020) and MYCOBANK Database (https://www.mycobank.org/page/Basic%20names%20search accessed on 20 October 2020).

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
