# Peer review of "Phylogenetic Review of Acaulospora (Diversisporales, Glomeromycota) and the Homoplasic Nature of Its Ornamentations"

_jof, 2022, doi:10.3390/jof8090892_

Round 1

Reviewer 1 Report

Reviewed paper deal with the diversity of very dynamic (from the nomenclatoric point of view) genus Acaulospora. The paper is meant to be review, however it brings it´s own phylogenetic analyses based on already published and deposited sequences. Quite strict rules to inclusion GenBank sequences (only species-identified....) does not correspond with list of used material where 3 sequences per species are listed. In the alignments, there is a lot of dubious positions which were further coded as gaps. This really concerns me in terms of the quality of the conducted analysis. I highly recommend to remove dubious position with some software (e.g. gBlocks) and redo analyses. Also, I am missing at least one analysis of all available sequences showing true diversity of Acaulospora and indicating real phylogenetic relationships.

The paper is hard to read. Authors did not pay attention towards proper use of nomenclatoric terms, which makes their explanations quite confusing (see further comments). Formating of the paper should be deeply revised e.g. names within lines 119-129 should be italicized; authors names does not follow widely accepted standards. Author´s name in the parenthesis means that name is not currently used and after parenthesis author of the actual name should be listed. English should be improved dramaticaly. Sentences with length of 5 lines should be ommitted.

Table 3 is confusing. Word "synonymy" in header suggest that those species in that column are already synonyms of Acaulospora. Also the sings used before species names indicate the same. In real, it is other way round. Also, header stated that you are evaluating species transferred from Acaulospora, in that case there should not be case of Kuklospora colombiana

Table 5 should be improved significantly, as the identificator some specimen voucher should be used instead of GenBank accession numbers

Figure 1 - drawings looked artificiall and unrealistic. I highly recommend to supplement your drawings with real microscopic pictures

Author Response

Response to Reviewer 1 Comments

Dear revisor, thank you for your comments.

All the changes made are highlighted in yellow in the word and pdf manuscript.

Please see below the answer for each point.

Point 1: The paper is meant to be review, however it brings it´s own phylogenetic analyses based on already published and deposited sequences.

Response 1: This manuscript is actually a review. It is indeed not just a descriptive article, since it presents some analysis. There are many reviews in which authors analyze the available data. For example, the article “Outline of Fungi and fungus-like taxa” by Wijayawardene et al. (2021), used the available database to review fungal clades. Some methodology was relocated to figure legend to make the text more concise.

Point 2: Quite strict rules to inclusion GenBank sequences (only species-identified....) does not correspond with list of used material where 3 sequences per species are listed.

Response 2: Sequences from uncultured or not morphologically described species were not included in our phylogenetic analysis, since our objective was to compare the phylogenetic relationships of the already described species with the ornamentation distribution in Acaulospora genus, as written in lines 306-310. Concerning the 3 sequences used, they did not appear in the trees because we collapsed the branches, but now we change the tree figures showing all the branches.

Point 3: In the alignments, there is a lot of dubious positions which were further coded as gaps. This really concerns me in terms of the quality of the conducted analysis. I highly recommend to remove dubious position with some software (e.g. gBlocks) and redo analyses. 

Response 3: We have tested many alignment programs, and the mafft was the best. Concerning the gaps, we agreed with the revisor and removed the gap as a fifth base and performed the analysis again. Please see the new alignment and trees.

Point 4: I am missing at least one analysis of all available sequences showing true diversity of Acaulospora and indicating real phylogenetic relationships.

Response 4: As mentioned in the item 4 of the manuscript. Sequences from uncultured or not morphologically described species were not included in our phylogenetic analysis, since our objective was to compare the phylogenetic relationships of the already described species with the ornamentation distribution in Acaulospora. So, we underline that the analysis aims to verify the phylogeny of Acaulospora according to the sequences deposited as such taxon. Besides most of these sequences from non-described Acaulospora species are from the ITS region alone, which, according to our data would not be the best choice for phylogenetic inference. We used all available sequences for described Acaulospora species in the concatenated analysis, however the phylogeny resolution was not the best.

Point 5:  Formating of the paper should be deeply revised e.g. names within lines 119-129 should be italicized; authors names does not follow widely accepted standards. Author´s name in the parenthesis means that name is not currently used and after parenthesis author of the actual name should be listed.

Response 5: The manuscript was reviewed and these mistakes were corrected accordingly.

Point 6: English should be improved dramaticaly. Sentences with length of 5 lines should be ommitted

Response 6: A native scientist speaker, Christian Joseph David Dore have kindly reviewed the English of our manuscript and we added him in the acknowledgments.

Point 7: Table 3 is confusing. Word "synonymy" in header suggest that those species in that column are already synonyms of Acaulospora. Also the sings used before species names indicate the same. In real, it is other way round. Also, header stated that you are evaluating species transferred from Acaulospora, in that case there should not be case of Kuklospora colombiana

Response 7: Table 3 was restructured and the mistakes were corrected.

Point 8: Table 5 should be improved significantly, as the identificator some specimen voucher should be used instead of GenBank accession numbers

Response 8: The specimen voucher identificatory is not available for many sequences and besides that our objective was to compare the phylogenetic relationships using only the sequences available. The slides with original specimen were not revisited.

Point 9: Figure 1 - drawings looked artificiall and unrealistic. I highly recommend to supplement your drawings with real microscopic pictures

Response 9: On the side of each drawing, a real picture of the ornamentations was included.

Reviewer 2 Report

This work provides a review focused on the phylogenetic analysis for species discrimination and the ornamentation patterns of the species in the genus Acaulospora, which belongs to Glomeromycota. However, the objectives for the work are not very clear, and it is not well stated for the background why the authors pose these questions. The manuscript is shaped like a report for literatures, but not a concise and broad review. I suggest that: first you should introduce the reason why you do these reviews and why it is important for the research of Acaulospora; secondly, in each section, it’s better to remove the content which could be omitted, and make the manuscript more concise and easy to read; thirdly, the text is with some grammar and linguistic mistakes and needs to be corrected throughout by a professional scientific writer and check again carefully.

Author Response

Response to Reviewer 2 Comments

 Dear revisor, thank you for your comments.

All the changes made are highlighted in yellow in the word and pdf manuscript.

Please see below the answer for each point.

Point 1: the objectives for the work are not very clear, and it is not well stated for the background why the authors pose these questions.

Response 1: We clarified better the aims of this work in the abstract by writing: The present review aimed to: (i) understand the evolutionary meaning of their different spore wall ornamentations, (ii) to define the best molecular marker for phylogenetic inferences, (iii) to address some specific issues, concerning the polyphyletic nature of Acaulospora lacunosa and Acaulospora scrobiculata, and the inclusion of Kuklospora species and (iv) to update the global geographical distribution of Acaulospora species. As such, the wall ornamentation of previously described Acaulospora species was reviewed and phylogenetic analyses, based on ITS and SSU-ITS-LSU (nrDNA) were carried out. Moreover, the already available type material of A. sporocarpia was inspected.

Point 2: The manuscript is shaped like a report for literatures, but not a concise and broad review. I suggest that: first you should introduce the reason why you do these reviews and why it is important for the research of Acaulospora; secondly, in each section, it’s better to remove the content which could be omitted, and make the manuscript more concise and easy to read

Response 2: We tried to better justify the review by doing small changes in the last two paragraphs of the introduction, lines 62-75, where we informed to the reader the potential application of Acaulospora for environmental recovery and agricultural activities. Also, we made an extensive review to make sentences more direct and corrected English mistakes. We cannot make this review more concise because it presents some analysis of the available data that we must detailed. We relocated some “methodology;’ to the phylogenies legends to make the reading easier.

Point 3:  the text is with some grammar and linguistic mistakes and needs to be corrected throughout by a professional scientific writer and check again carefully.

Response 3: A native scientist speaker, Christian Joseph David Dore have kindly reviewed the English of our manuscript and we added him in the acknowledgments.

Reviewer 3 Report

Please see the comments in attached file.

Author Response

Response to Reviewer 3 Comments

Dear revisor, thank you for your comments.

All the changes made are highlighted in yellow in the word and pdf manuscript.

Please see below the answer for each point.

Point1: minor corrections in pdf

- the bold has been removed from figure legends;

- we corrected all the references that were mistakenly cited in the text in lines: 59, 89, 92, 232, 239, 242, 246, 252, 255, 258, 304, 390, 393, 399-401;

- we corrected the references in table 3 and 4;

- all taxon names were italicized.

Reviewer 4 Report

The paper entitled “Phylogenetic review of Acaulospora (Diversisporales, Glomeromycota) and the homoplasic nature of its ornamentations” by da Silva et al. aimed to update the global geographical distribution of Acaulospora species, inspected available type material of A. sporocarpia, investigate the ornamentation patterns of Acaulospora species and construct phylogenetic analysis, based on multi-genes. This study is important to help researchers deeper understand the Acaulospora fungi. The ‘Materials and methods’ section is missing in here. How the authors obtained these wonderful maps and other illustrations? I think the article is valuable, but I have some minor comments (see attached PDF).

Author Response

Response to Reviewer 4 Comments

Dear revisor, thank you for your comments.

All the changes made are highlighted in yellow in the word and pdf manuscript.

Please see below the answer for each point.

Point 1: The ‘Materials and methods’ section is missing in here

Response 1: This manuscript is actually a review. It is indeed not just a descriptive article, since it presents some analysis. There are many reviews in which authors analyse the available data. For example, the article “Outline of Fungi and fungus-like taxa” by Wijayawardene et al. (2021), used the available database to review fungal clades. Some methodology was relocated to figure legend to make the text more concise.

Point 2:  How the authors obtained these wonderful maps and other illustrations?

 Response 2: all the illustrations were made or edited using inkscape.

Point 3: minor corrections in pdf

- the bold was removed from the word the in the abstract;

- we specified that ornamentations are actually spore wall ornamentations in the abstract and in all other places in the text (lines 72, 122, 135, 422, 443, 510);

- We added an example of shared morphological characteristic between different species (lines 51-52);

- We added references in table 5;

- We corrected all the references that were mistakenly cited in the text in lines: 59, 89, 92, 232, 239, 242, 246, 252, 255, 258, 304, 390, 393, 399-401;

- Kuklospora kentinensis was replaced with Acaulospora kentinensis in table 2;

- Kuklospora spinosa was not replaced with Acaulospora spinosa because as explained in lines 406-412 there is a homonym problem. Also this species does not have available sequences in databases yet;

- no reference was cited for “the most suitable molecular marker available to understand the evolution of Acaulospora” in lines 203-204 because it was our own conclusion after analyzing the data;

- nov/2018 and dez/2020 was corrected to November 2018 and December 2020 in line 210-211;

- Acaulospora was italicized in the legend of figure 2;

- Concerning the sequences of Kuklospora colombiana and Kuklospora kentinensis, we considered trustable sequences. We used the same sequences used by other authors (ref 48, 128, 67, 70, 91, 74, 53).

- Concerning the question about whether the ornamentations are only for soil-based habitats, we did not understand exactly the revisor concern. Does the question is about the relation between ornamentations and plant hosts? If so, there is nothing in literature indicating that the species with a certain ornamentation are more related to a plant host or even to a certain soil habitat.

- Examples of additional genes, for molecular markers, were included in line 498.

Round 2

Reviewer 1 Report

Authors did good job in order to improve reviewed manuscript. They addressed my major concern in terms of conducted analyses. However, I feel manuscript still need thorough revision of mistakes and typos. I mentioned only few of them together with some parts which should be improved once again

e.g. References no. 33 should have not abbreviated journal name, many references are not provided with DOI numbers e.g. 32

Table 3 - I suggest heading Overview instead of Synonymous, in the header replace synonymy for Original name since they may contain basionyms. Also, it is still not clear why you include in this table Acaulospora colombiana, A. kentinensis and A. brasiliensis, when you declare that table contain species currently not belonging to Acaulospora. Additionally, are you sure that names K. kentinensis and K. colombiana are based on different types than currently used names?

Figure 3 - please explain what are that color indication for each clade

Figure 4 - First sentence - ITS tree using partial ITS. second sentence the complete ITS1-5.8S-ITS2 was used. Also improve the graphics, make vertical lines thicker in order to make whole topology easily visible

Author Response

Response to Reviewer 1 Comments

Dear revisor, thank you for your comments.

All the changes made are highlighted in green in the word and pdf manuscript. Please see below the answer for each point.

Point 1: Authors did good job in order to improve reviewed manuscript. They addressed my major concern in terms of conducted analyses. However, I feel manuscript still need thorough revision of mistakes and typos. I mentioned only few of them together with some parts which should be improved once again.

Response 1: Thank you for kind answer. We re-wrote some parts of the text in an attempt to make it clearer.

Point 2: e.g. References no. 33 should have not abbreviated journal name, many references are not provided with DOI numbers e.g. 32

Response 2: We correct the reference number 33 and provided doi number for all references for which it is available.

Point 3: Table 3 - I suggest heading Overview instead of Synonymous, in the header replace synonymy for Original name since they may contain basionyms. Also, it is still not clear why you include in this table Acaulospora colombiana, A. kentinensis and A. brasiliensis, when you declare that table contain species currently not belonging to Acaulospora. Additionally, are you sure that names K. kentinensis and K. colombiana are based on different types than currently used names?

Response 3: Thank you for suggestion. We change the table 3 as requested. We removed K. colombiana e K. kentinensis but we still maintained A. brasiliensis. A. brasiliensis sequences were obtained from an isolate from Scotland, while the original material was described in Brazil. The Brazilian and Scottish isolates present some remarkable morphological differences. However there are no DNA sequences from the Brazilian isolate available to confirm its phylogenetic position.

Point 4: Figure 3 - please explain what are that color indication for each clade

Response 4: The color indicate the type of spore wall ornamentation in the clade. We added this information in the legend.

Point 5: Figure 4 - First sentence - ITS tree using partial ITS. second sentence the complete ITS1-5.8S-ITS2 was used. Also improve the graphics, make vertical lines thicker in order to make whole topology easily visible

Response 5: Images are best viewed by changing the word settings. I changed the resolution of the file and I hope you can see the images better now.

Reviewer 2 Report

Please check carefully again throughout the manuscript. There are still some minor mistakes.

Author Response

Response to Reviewer 2 Comments

Dear revisor, thank you for your comments.

Minor revision

All comments have been checked and corrections made. All the changes made are highlighted in green in the word and pdf manuscript.